# Ditto: A Simple and Efficient Approach to Improve Sentence Embeddings

**Qian Chen, Wen Wang, Qinglin Zhang, Siqi Zheng**
**Chong Deng, Hai Yu, Jiaqing Liu, Yukun Ma, Chong Zhang**
Speech Lab, Alibaba Group
{tanqing.cq,w.wang}@alibaba-inc.com

## Abstract

Prior studies diagnose the anisotropy problem in sentence representations from pre-trained language models, e.g., BERT, without fine-tuning. Our analysis reveals that the sentence embeddings from BERT suffer from a bias towards uninformative words, limiting the performance in semantic textual similarity (STS) tasks. To address this bias, we propose a simple and efficient unsupervised approach, **Di**agonal **Att**ention Po**o**ling (**Ditto**), which weights words with model-based importance estimations and computes the weighted average of word representations from pre-trained models as sentence embeddings. Ditto can be easily applied to any pre-trained language model as a postprocessing operation. Compared to prior sentence embedding approaches, Ditto does not add parameters nor requires any learning. Empirical evaluations demonstrate that our proposed Ditto can alleviate the anisotropy problem and improve various pre-trained models on the STS benchmarks.[1]

## 1 Introduction

Pre-trained language models (PLM) such as BERT (Devlin et al., 2019), RoBERTa (Liu et al., 2019), and ELECTRA (Clark et al., 2020) have achieved great success in a wide variety of natural language processing tasks. However, Reimers and Gurevych (2019) finds that sentence embeddings from the original BERT underperform traditional methods such as GloVe (Pennington et al., 2014). Typically, an input sentence is first embedded by the BERT embedding layer, which consists of token embeddings, segment embeddings, and position embeddings. The output is then encoded by the Transformer encoder and the hidden states at the last layer are averaged to obtain the sentence embeddings. Prior studies identify the anisotropy problem as a critical factor that harms

BERT-based sentence embeddings, as sentence embeddings from the original BERT yield a high similarity between any sentence pair due to the narrow cone of learned embeddings (Li et al., 2020).

Prior approaches for improving sentence embeddings from PLMs fall into three categories. The first category of approaches *does not require any learning (that is, learning-free)*. Jiang et al. (2022) argues that the anisotropy problem may be mainly due to the static token embedding bias, such as token frequency and case sensitivity. To address these biases, they propose the *static remove biases avg.* method which removes top-frequency tokens, subword tokens, uppercase tokens, and punctuations, and uses the average of the remaining token embeddings as sentence representation. However, this approach does not use the contextualized representations of BERT and may not be effective for short sentences as it may exclude informative words. The prompt-based method (*last manual prompt*) (Jiang et al., 2022) uses a template to generate sentence embeddings. An example template is *This sentence: "[X]" means [MASK] .*, where [X] denotes the original sentence and the last hidden states in the [MASK] position are taken as sentence embeddings. However, this method has several drawbacks. (1) It increases the input lengths, which raises the computation cost. (2) It relies on using the [MASK] token to obtain the sentence representation, hence unsuitable for PLMs not using [MASK] tokens (e.g., ELECTRA). (3) The performance heavily depends on the quality of manual prompts which relies on human expertise (alternatively, OptiPrompt (Zhong et al., 2021) requires additional unsupervised contrastive learning).

The second category of approaches *fixes the parameters of the PLM* and improves sentence embeddings through post-processing methods that *require extra learning*. BERT-flow (Li et al., 2020) addresses the anisotropy problem by introducing a flow-based generative model that transforms

---

[1]The source code can be found at https://github.com/alibaba-damo-academy/SpokenNLP/tree/main/ditto.

the BERT sentence embedding distribution into a smooth and isotropic Gaussian distribution. BERT-whitening (Su et al., 2021) uses a whitening operation to enhance the isotropy of sentence representations. Both BERT-flow and BERT-whitening require Natural Language Inference (NLI)/Semantic Textual Similarity (STS) datasets to train the flow network or estimate the mean values and covariance matrices as the whitening parameters.

The third category *updates parameters of the PLM by fine-tuning or continually pre-training the PLM using supervised or unsupervised learning*, which is computationally intensive. For example, Sentence-BERT (SBERT) (Reimers and Gurevych, 2019) fine-tunes BERT using a siamese/triplet network on NLI and STS datasets. SimCSE (Gao et al., 2021) explores contrastive learning. Unsupervised SimCSE uses the same sentences with different dropouts as positives and other sentences as negatives, and supervised SimCSE explores NLI datasets and treats entailment pairs as positives and contradiction pairs as hard negatives.

In this work, we first analyze BERT sentence embeddings. (1) We use a parameter-free probing method to analyze BERT and Sentence-BERT (SBERT) (Reimers and Gurevych, 2019) and find that the compositionality of informative words is crucial for generating high-quality sentence embeddings as from SBERT. (2) Visualization of BERT attention weights reveals that certain self-attention heads in BERT are related to informative words, specifically self-attention from a word to itself (that is, the diagonal values of the attention matrix). Based on these findings, we propose a simple and efficient approach, **Di**agonal At**t**ention Po**o**ling (**Ditto**), to improve sentence embeddings from PLM without requiring any learning (that is, *Ditto is a learning-free method*). We find that Ditto improves various PLMs and strong sentence embedding methods on STS benchmarks.

## 2   Analyze BERT Sentence Embeddings

**Observation 1: The compositionality of informative words is crucial for high-quality sentence embeddings.** Perturbed masking (Wu et al., 2020) is a parameter-free probing technique for analyzing PLMs (e.g., BERT). Given a sentence $\mathbf{x} = [x_1, x_2, \ldots, x_N]$, perturbed masking applies a two-stage perturbation process on each pair of tokens $(x_i, x_j)$ to measure the impact that a token $x_j$ has on predicting the other token $x_i$. Details of perturbed masking can be found in Ap-

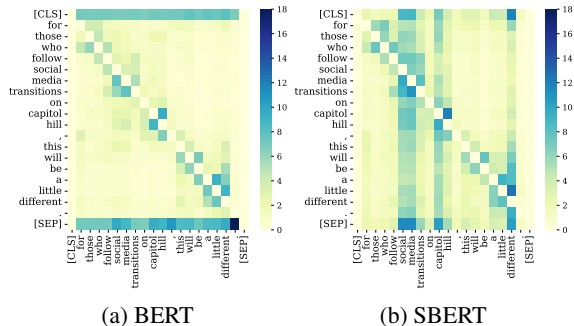

(a) BERT          (b) SBERT

Figure 1: The heatmap shows the impact matrix for the sentence "For those who follow social media transitions on Capitol Hill, this will be a little different.". The impact matrices are computed using BERT (bert-base-uncased) and SBERT (bert-base-nli-stsb-mean-tokens) on Hugging Face, respectively.

pendix A.1. Prior works use perturbed masking to recover syntactic trees from BERT (Wu et al., 2020). Different from prior works, we use perturbed masking to analyze the original BERT and a strong sentence embedding model, supervised Sentence-BERT (SBERT) (Reimers and Gurevych, 2019). Figure 1 shows the heatmap representing the impact matrix $\mathcal{F}$ for an example sentence in the English PUD treebank (Zeman et al., 2017). The y-axis represents $x_i$ and the x-axis represents $x_j$. A higher value in $\mathcal{F}_{ij}$ indicates that a word $x_j$ has a greater impact on predicting another word $x_i$. Comparing the impact matrices of BERT and SBERT, we observe that the impact matrix of SBERT exhibits prominent vertical lines on informative tokens such as "social media", "Capitol Hill", and "different", which implies that informative tokens have a high impact on predicting other tokens, hence masking informative tokens could severely affect predictions of other tokens in the sentence. In contrast, BERT does not show this pattern. This observation implies that the compositionality of informative tokens could be a strong indicator of high-quality sentence embeddings of SBERT. Furthermore, we compute the correlations between the impact matrix and TF-IDF (Sparck Jones, 1972) which measures the importance of a word, and report results in Table 3. We find that the impact matrix of SBERT has a much higher correlation with TF-IDF than the impact matrix of BERT, which is consistent with the observation above. Notably, ELECTRA performs poorly on STS tasks and shows a weak correlation with TF-IDF. Consequently, we hypothesize that sentence embeddings of the original BERT and ELECTRA may be bi-

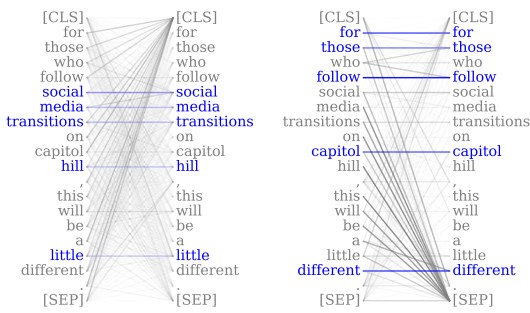 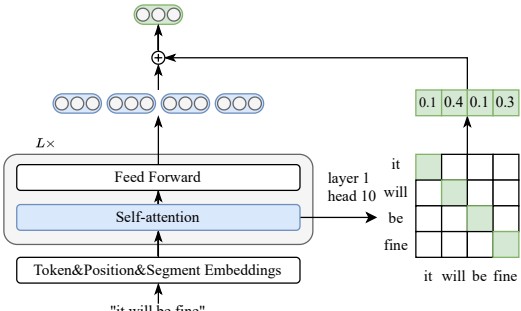

Figure 2: Illustration of attention weights for BERT head 1-10 (on the left) and head 11-11 (on the right). The darkness of a line represents the value of the attention weights. The top-5 diagonal values of the attention matrix are colored blue.

Figure 3: The diagram of our proposed diagonal attention pooling (Ditto) method.

ased towards uninformative words, hence limiting their performance on STS tasks.

**Observation 2: Certain self-attention heads of BERT correspond to word importance.** Although SBERT has a higher correlation with TF-IDF than BERT as verified in Observation 1, BERT still shows a moderate correlation. Thus, we hypothesize that the semantic information of informative words is already encoded in BERT but has yet to be fully exploited. Prior research (Clark et al., 2019) analyzes the attention mechanisms of BERT by treating each attention head as a simple, no-training-required classifier that, given a word as input, outputs the other word that it most attends to. Certain attention heads are found to correspond well to linguistic notions of syntax and coreference. For instance, heads that attend to the direct objects of verbs, determiners of nouns, objects of prepositions, and coreferent mentions are found to have remarkably high accuracy. We believe that the attention information in BERT needs to be further exploited. We denote a particular attention head by <layer>-<head number> (l-h), where for a BERT-base-sized model, *layer* ranges from 1 to 12 and *head number* ranges from 1 to 12. We visualize the attention weights of each head in each layer of BERT and focus on informative words. We then discover that self-attention from a word to itself (that is, the diagonal value of the attention matrix, named **diagonal attention**) of certain heads may be related to the importance of the word. As shown in Figure 2, the informative words "social media transitions", "hill", and "little" have high diagonal values of the attention matrix of head 1-10. Section 4 will demonstrate that diagonal attentions indeed have a strong correlation with TF-IDF weights.

# 3 Diagonal Attention Pooling

Inspired by the two observations in Section 2, we propose a novel learning-free method called **Di**agonal At**t**ention P**o**oling (**Ditto**) to improve sentence embeddings for PLMs, illustrated in Figure 3. Taking BERT as an example, the input to the Transformer encoder is denoted as $\mathbf{h}^0 = [h_1^0, \ldots, h_N^0]$, and the hidden states at each Transformer encoder layer are denoted as $\mathbf{h}^l = [h_1^l, \ldots, h_N^l], l \in \{1, \ldots, L\}$. Typically, the hidden states of the last layer of the PLM are averaged to obtain the fixed-size sentence embeddings, as $\frac{1}{N}\sum_{i=1}^{N} h_i^L$ (denoted as *last avg.*). Alternatively, we can also average the static word embeddings $\frac{1}{N}\sum_{i=1}^{N} h_i^0$ (denoted as *static avg.*), or average the hidden states from the first and last layers $\frac{1}{2N}\sum_{i=1}^{N} (h_i^0 + h_i^L)$ (denoted as *first-last*) to obtain the sentence embeddings. Ditto weights the hidden states with diagonal attention of a certain head. For example, to obtain the sentence embeddings from the first-last hidden states of BERT using Ditto, we first obtain the diagonal values $[\mathcal{A}_{11}, \ldots, \mathcal{A}_{NN}]$ of the attention matrix $\mathcal{A}$ for head $l$-$h$ of BERT, where $l$ and $h$ are treated as hyperparameters and are optimized on a development set based on the STS performance. Then, we compute $\frac{1}{2}\sum_{i=1}^{N} \mathcal{A}_{ii}(h_i^0 + h_i^L)$ as the sentence embeddings[2]. Note that the impact matrix (Section 2) correlates well with TF-IDF (as shown in Table 3) and hence may also improve sentence embeddings. However, the learning-free Ditto is much more efficient than computing the impact matrix, which is computationally expensive.

# 4 Experiments and Analysis

Following prior works (Jiang et al., 2022; Li et al., 2020; Su et al., 2021; Gao et al., 2021), we exper-

---

[2]We did not normalize with $N$ since $\mathcal{A}_{ii} < 1$ and normalization by $N$ may result in very small values.

Table 1: Performance of different sentence embedding methods on STS tasks (as Average Spearman's correlation). Table 6 in the Appendix reports detailed results.

| Method | Avg. |
|---|---|
| *Learning-free methods* | |
| BERT static avg. | 56.02 |
| BERT last avg. | 52.57 |
| BERT first-last avg. | 56.70 |
| BERT static remove biases avg. | 63.10 |
| BERT last manual prompt | **67.85** |
| BERT static **Ditto** (**Ours**) | 61.77 |
| BERT last **Ditto** (**Ours**) | 59.07 |
| BERT first-last **Ditto** (**Ours**) | 64.77 |
| *Methods that fix BERT parameters but require extra learning* | |
| BERT-flow | 66.55 |
| BERT-whitening | 66.28 |
| BERT last manual and continuous prompt | **73.59** |
| BERT first-last TF-IDF (**Ours**) | 65.45 |
| *Methods that update BERT parameters* | |
| Unsup. BERT SimCSE | 76.25 |
| Sup. BERT SimCSE | 81.57 |
| Sup. SBERT first-last avg. | 84.94 |
| Sup. SBERT first-last **Ditto** (**Ours**) | **85.11** |

Table 2: Sentence embedding performance of Ditto and learning-free baselines on PLMs, measured by average Spearman's correlation on the test sets of 7 STS tasks.

| Method | BERT | RoBERTa | ELECTRA |
|---|---|---|---|
| First-last avg. | 56.70 | 56.57 | 36.28 |
| Last manual prompt | **67.85** | 61.08 | 19.44 |
| First-last **Ditto** (**Ours**) | 64.77 | **61.96** | **52.00** |

iment on the 7 common STS datasets, the widely used benchmark for evaluating sentence embeddings. Appendix A.2 presents dataset and implementation details.

**Main Results**    The first group of Table 1 presents the results of the three common learning-free methods (Jiang et al., 2022) described in Section 3, including *static avg.*, *last avg.*, and *first-last avg.*, and two learning-free baselines described in Section 1, *static remove biases avg.* and *last manual prompt* (Jiang et al., 2022). Although *last manual prompt* achieves 67.85, different templates can have a significant impact on its performance, ranging from 39.34 to 73.44 on STS-B dev set (Jiang et al., 2022). Applying Ditto to *static avg.*, *last avg.*, and *first-last avg.* achieves absolute gains of **+5.75**, **+6.50**, and **+8.07** on the Avg. score, respectively.[3]

The second group of Table 1 presents results from methods that fix BERT parameters but require

---

[3]Since *static remove biases avg.* may remove important tokens and *last manual prompt* uses the last hidden states of [MASK] as sentence embeddings instead of average pooling, they are not suitable for applying Ditto.

extra learning. Note that our learning-free *BERT first-last Ditto* achieves comparable performance to BERT-flow and BERT-whitening in this group. For further analyzing Ditto, we compute TF-IDF weights on $10^6$ sentences randomly sampled from English Wikipedia as token importance weights and use the weighted average of the first-last hidden states as sentence embeddings, denoted as *first-last TF-IDF* (the 4th row in this group). *First-last TF-IDF* yields **+8.75** absolute gain over the *first-last avg.* baseline, only slightly better than the +8.07 absolute gain from our learning-free Ditto (with $l$ and $h$ searched on only 1500 samples).

The third group of Table 1 presents results of strong baselines that update BERT parameters through unsupervised or supervised learning, including unsupervised SimCSE (*Unsup. BERT SimCSE*), supervised SimCSE (*Sup. BERT SimCSE*), and supervised SBERT. We find that applying Ditto on the highly competitive supervised learning method *Sup. SBERT first-last avg.* still achieves an absolute gain of 0.17 (84.94→85.11), demonstrating that Ditto could also improve strong supervised sentence embedding methods. Note that since SimCSE uses the [CLS] representation as sentence embeddings instead of average pooling, SimCSE is not suitable for applying Ditto.

**Effectiveness of Ditto on Different PLMs**    Table 2 compares the baselines *first-last avg.* and *last manual prompt* and our *first-last Ditto* method on different PLMs. Note that *last manual prompt* does not work for ELECTRA because this method relies on using the [MASK] token as the sentence embeddings while the ELECTRA discriminator is trained without [MASK] tokens. *Ditto* consistently works well on ELECTRA and greatly outperforms the two baselines on RoBERTa and ELECTRA, while underperforming *last manual prompt* on BERT.

Table 3: Correlation of the impact matrix with the TF-IDF weights. Column **STS avg.** shows the average Spearman's correlation on the test sets of 7 STS tasks. Columns **Pear.** and **Spear.** show Pearson's correlation and Spearman's correlation between the mean values of the impact matrix $\frac{1}{N}\sum_{i=1}^{N}\mathcal{F}_{ij}$ and the TF-IDF weights for 1K sentences randomly sampled from the English PUD treebank.

| Method | STS avg. | Pear. | Spear. |
|---|---|---|---|
| BERT | 56.70 | 57.27 | 57.44 |
| SBERT | 84.94 | 62.90 | 70.21 |
| ELECTRA | 36.28 | 12.97 | 21.91 |

**Correlation with TF-IDF** To further analyze correlations between diagonal attentions and word importance, we select the 4 heads corresponding to the Top-4 Ditto performance based on Spearman's correlation on the STS-B development set, and compute correlations between diagonal values of the self-attention matrix of these heads and TF-IDF weights. Table 4 shows all Top-4 heads exhibit moderate or strong correlations with TF-IDF weights. We find high-performing heads are usually in the bottom layers (Section A.2), which is consistent with the findings in Clark et al. (2019) that the bottom layer heads broadly attend to the entire sentence.

Table 4: The sentence embedding performance of **BERT first-last Ditto** using different attention heads and the correlation with the TF-IDF weights. **Dev** denotes Spearman's correlation on the STS-B development set. **Test** denotes the average Spearman's correlation on the test sets of 7 STS tasks. **Pear.** and **Spear.** denote Pearson's correlation and Spearman's correlation between the diagonal values of the attention matrix for the certain attention head and the TF-IDF weights on sentences in the STS-B task.

| Method | Dev | Test | Pear. | Spear. |
|---|---|---|---|---|
| Ditto Head 1-10 | **74.56** | 64.77 | 64.34 | 63.56 |
| Ditto Head 2-12 | 73.13 | **65.00** | 47.30 | 44.17 |
| Ditto Head 11-11 | 70.59 | 62.46 | 47.64 | 44.68 |
| Ditto Head 1-7 | 69.54 | 60.65 | **65.98** | **64.30** |

**Uniformity and Alignment** We use the analysis tool from prior works (Gao et al., 2021) to evaluate the quality of sentence embeddings by measuring the alignment between semantically related positive pairs and uniformity of the whole representation space. Gao et al. (2021) finds that sentence embedding models with better alignment and uniformity generally achieve better performance. Figure 4 shows the uniformity and alignment of different sentence embedding models along with their averaged STS results. Lower values indicate better alignment and uniformity. We find that Ditto improves uniformity at the cost of alignment degradation for all PLMs, similar to the flow and whitening methods as reported in Gao et al. (2021). Compared to Ditto, flow and whitening methods achieve larger improvements in uniformity but also cause larger degradations in alignment.

**Cosine Similarity** We use the cosine similarity metric from Ethayarajh (2019) to measure the isotropy of sentence representations. Isotropy means that the word or sentence representations

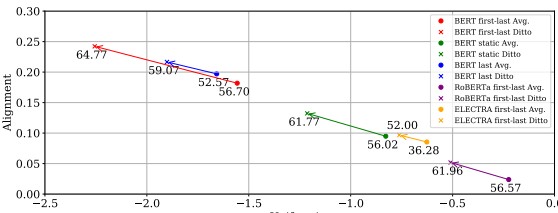

Figure 4: Alignment-uniformity plot of baselines and Ditto on different PLMs. The arrow indicates the changes. For both alignment and uniformity, smaller numbers are better.

are directionally uniform, and the average cosine similarity between random samples should be close to zero. Ethayarajh (2019) originally applied this metric to word representations, and we adapt it to sentence representations in our study. We sample 1000 sentences from the English Wikipedia dataset and compute the average cosine similarity of their representations. Table 5 shows the results. Lower values indicate better isotropy. Our proposed Ditto method improves the isotropy of all three learning-free baselines: static avg., last avg., and first-last avg. This result is consistent with the uniformity analysis in Figure 4, where Ditto also enhances the uniformity of different sentence embedding models.

Table 5: The average cosine similarity of sentence representations for Ditto and learning-free baselines.

| Method | avg. Cosine Similarity |
|---|---|
| BERT static avg. | 0.843 |
| BERT static Ditto | 0.768 |
| BERT last avg. | 0.508 |
| BERT last Ditto | 0.458 |
| BERT first-last avg. | 0.566 |
| BERT first-last Ditto | 0.403 |

## 5 Conclusions

We propose a simple and learning-free Diagonal Attention Pooling (Ditto) approach to address the bias towards uninformative words in BERT sentence embeddings. Ditto weights words with model-based importance estimations and can be easily applied to various PLMs. Experiments show that Ditto alleviates the anisotropy problem and improves strong sentence embedding baselines.

## Limitations

Although our proposed simple and learning-free Ditto approach demonstrates effectiveness in allevi-

ating the anisotropy problem and improving strong sentence embedding baselines, there are several limitations. Firstly, we conduct our experiments solely on the English sentence embedding models and the English Semantic Textual Similarity (STS) datasets. We hypothesize that the two observations in Section 2 will hold true on pre-trained models for other languages, hence we predict that Ditto, which is based on the two observations, will be effective in improving sentence embeddings for languages other than English. We plan to investigate the efficacy of Ditto on improving sentence embeddings for other languages in future work. Secondly, while we select the attention head (that is, determining $l$ and $h$) by conducting a grid search of all attention heads based on the performance of the STS development set, we will explore other approaches for selecting attention heads for Ditto in future studies. Lastly, we focus on using Semantic Textual Similarity tasks for evaluating sentence embeddings in this work. We plan to investigate the quality of sentence embeddings in more tasks, such as information retrieval.

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

## A Appendix

### A.1 Details of Perturbed Masking

In the first stage, we replace $x_i$ with the [MASK] token, resulting in a new sequence $\mathbf{x}\backslash\{x_i\}$. The representation of this sequence is denoted as $H(\mathbf{x}\backslash\{x_i\})_i$. In the second stage, we mask out $x_j$ in addition to $x_i$ to obtain the second corrupted sequence $\mathbf{x}\backslash\{x_i, x_j\}$. The representation of this sequence is denoted as $H(\mathbf{x}\backslash\{x_i, x_j\})_i$. Thus we obtain an impact matrix $\mathcal{F} \in \mathbb{R}^{N \times N}$ by computing the Euclidean distance between the two representations $\mathcal{F}_{ij} = d(H(\mathbf{x}\backslash\{x_i\})_i, H(\mathbf{x}\backslash\{x_i, x_j\})_i)$.

### A.2 Dataset and Implementation Details

We conduct experiments on 7 common STS datasets, namely, STS tasks 2012-2016 (Agirre et al., 2012, 2013, 2014, 2015, 2016), STS-B (Cer et al., 2017), and SICK-R (Marelli et al., 2014), following prior works. These 7 STS datasets are widely used benchmarks for evaluating sentence embeddings. Each dataset consists of sentence pairs scored from 0 to 5 to indicate the semantic similarity. For evaluation, we follow the setting of Reimers and Gurevych (2019) and report the average Spearman's correlation on the test sets of all 7 STS tasks (that is, the "all" setting), without using an additional regressor. Our implementation is based on the SimCSE GitHub repository[4] and we modify it to fit our purposes. We conduct a grid search of the attention head $l$-$h$ for Ditto based on Spearman's correlation on the STS-B development set (1500 samples). In this way, we select head 1-10 for BERT[5], head 1-5 for RoBERTa[6], head 1-11 for ELECTRA[7], and head 3-7 for SBERT[8]. The TF-IDF weights are learned on $10^6$ sentences randomly sampled from English Wikipedia[9] using the gensim tool[10]. We also utilize the English Wikipedia dataset and randomly sampled 1000 sentences to calculate the average cosine similarity of sentence representations. We conduct experiments using a single Tesla V100 GPU. Note that BERT-flow and BERT-whitening papers use the full target dataset (including all sentences in the train, development, and test sets, and excluding all labels) and optionally the NLI corpus (SNLI (Bowman et al., 2015) and MNLI corpus (Williams et al., 2018)) for training.

---

[4] https://github.com/princeton-nlp/SimCSE
[5] https://huggingface.co/bert-base-uncased
[6] https://huggingface.co/roberta-base
[7] https://huggingface.co/google/electra-base-discriminator
[8] https://huggingface.co/sentence-transformers/bert-base-nli-stsb-mean-tokens
[9] https://huggingface.co/datasets/princeton-nlp/datasets-for-simcse/resolve/main/wiki1m_for_simcse.txt
[10] https://radimrehurek.com/gensim/models/tfidfmodel.html

Table 6: The performance comparison of different sentence embedding methods on STS tasks (Spearman's correlation).

| Method | STS12 | STS13 | STS14 | STS15 | STS16 | STS-B | SICK-R | Avg. |
|---|---|---|---|---|---|---|---|---|
| *Learning-free methods* | | | | | | | | |
| BERT static avg. | 42.38 | 56.74 | 50.60 | 65.08 | 62.39 | 56.82 | 58.15 | 56.02 |
| BERT last avg. | 30.87 | 59.89 | 47.73 | 60.29 | 63.73 | 47.29 | 58.22 | 52.57 |
| BERT first-last avg. | 39.70 | 59.38 | 49.67 | 66.03 | 66.19 | 53.87 | 62.06 | 56.70 |
| BERT static remove biases avg. (Jiang et al., 2022) | 53.09 | 66.48 | 65.09 | 69.80 | 67.85 | 61.60 | 57.80 | 63.10 |
| BERT last manual prompt (Jiang et al., 2022) | 60.96 | 73.83 | 62.18 | 71.54 | 68.68 | 70.60 | 67.16 | **67.85** |
| BERT static **Ditto** (**Ours**) | 52.61 | 62.72 | 59.88 | 70.40 | 65.60 | 63.34 | 57.85 | 61.77 |
| BERT last **Ditto** (**Ours**) | 43.58 | 64.84 | 53.27 | 66.06 | 65.77 | 58.88 | 61.11 | 59.07 |
| BERT first-last **Ditto** (**Ours**) | 53.77 | 67.99 | 59.78 | 73.77 | 69.66 | 66.76 | 61.64 | **64.77** |
| *Methods that fix BERT parameters but require extra learning* | | | | | | | | |
| BERT-flow (Li et al., 2020) | 58.40 | 67.10 | 60.85 | 75.16 | 71.22 | 68.66 | 64.47 | 66.55 |
| BERT-whitening (Su et al., 2021) | 57.83 | 66.90 | 60.90 | 75.08 | 71.31 | 68.24 | 63.73 | 66.28 |
| BERT last manual and continuous prompt (Jiang et al., 2022) | 64.56 | 79.96 | 70.05 | 79.37 | 75.35 | 77.25 | 68.56 | **73.59** |
| BERT first-last TF-IDF (**Ours**) | 53.78 | 73.03 | 59.49 | 71.98 | 70.16 | 66.99 | 62.73 | 65.45 |
| *Methods that update BERT parameters (supervised or unsupervised learning)* | | | | | | | | |
| Unsup. BERT SimCSE (Gao et al., 2021) | 68.40 | 82.41 | 74.38 | 80.91 | 78.56 | 76.85 | 72.23 | 76.25 |
| Sup. BERT SimCSE (Gao et al., 2021) | 75.30 | 84.67 | 80.19 | 85.40 | 80.82 | 84.25 | 80.39 | 81.57 |
| Sup. SBERT first-last avg. (Reimers and Gurevych, 2019) | 80.67 | 88.00 | 89.78 | 90.28 | 82.12 | 85.16 | 78.55 | 84.94 |
| Sup. SBERT first-last **Ditto** (**Ours**) | 80.43 | 88.63 | 89.84 | 90.39 | 82.72 | 85.62 | 78.15 | **85.11** |