# OpenReview forum: "Ditto: A Simple and Efficient Approach to Improve Sentence Embeddings"
_EMNLP/2023/Conference — EMNLP 2023 Main_

### Official Review · Reviewer_Z8t9 · 2023-08-01

**Soundness:** 4

**Excitement:**

3: Ambivalent: It has merits (e.g., it reports state-of-the-art results, the idea is nice), but there are key weaknesses (e.g., it describes incremental work), and it can significantly benefit from another round of revision. However, I won't object to accepting it if my co-reviewers champion it.

**Paper Topic And Main Contributions:**

This paper proposes Diagonal Attention Pooling (Ditto), which weights words with model-based importance estimations and computes the weighted average of word representations from pre-trained models as sentence embeddings. The resulting embeddings show better results on STS benchmark against various baselines.

**Reasons To Accept:**

1. The summarization for the category of existing methods for improving sentence embeddings is nice.
2. The motivation of the proposed method is clear and the results are solid.

**Reasons To Reject:**

1. The conclusion for "alleviate the anisotropy problem" is not supported enough. Uniformity and alignment are just two metrics used by Gao et al. 2021. Some other methods can be seen in [1][2].

[1] All-but-the-top: Simple and effective postprocessing for word representations. ICLR 2018
[2] A Cluster-based Approach for Improving Isotropy in Contextual Embedding Space. ACL 2021

**Reproducibility:**

4: Could mostly reproduce the results, but there may be some variation because of sample variance or minor variations in their interpretation of the protocol or method.

**Reviewer Confidence:**

4: Quite sure. I tried to check the important points carefully. It's unlikely, though conceivable, that I missed something that should affect my ratings.

---

> ### Author Rebuttal · Authors · 2023-08-28
>
> We would like to express our gratitude to the reviewer for finding our summarization of existing methods nice and for considering our proposed method to be clear with solid results. We appreciate the constructive feedback. Below, we address all of your questions.
>
> ---
>
> **Q1: The conclusion for "alleviate the anisotropy problem" is not supported enough. Uniformity and alignment are just two metrics used by Gao et al. 2021. Some other methods can be seen in [1][2].
> [1] All-but-the-top: Simple and effective postprocessing for word representations. ICLR 2018 [2] A Cluster-based Approach for Improving Isotropy in Contextual Embedding Space. ACL 2021**
>
> **Response:**
> We appreciate the references you provided regarding metrics for the anisotropy problem. We have taken these works into consideration and found that they are not specifically designed for sentence embeddings.
>
> In "All-but-the-top: Simple and effective postprocessing for word representations" by Mu et al. and "A Cluster-based Approach for Improving Isotropy in Contextual Embedding Space" by Rajaee et al., the isotropy score measures the variance of the inner products between word vectors normalized by the square of the mean norm of all the word vectors. The higher the isotropy score, the more isotropic the word representations are. However, calculating the isotropy score requires obtaining the complete set of word vectors from a pre-trained model. This makes it infeasible to extend the isotropy score to evaluate sentence embeddings since enumerating all possible sentences and then computing their corresponding sentence embeddings are infeasible. Therefore, finding better metrics for evaluating the anisotropy problem of sentence embeddings remains an area for future research.
>
> Uniformity and alignment are commonly used metrics to analyze the quality of sentence embeddings (Line 296-301). For example, they have also been used in other works [1][2] in addition to (Gao et al., 2021).
>
> [1] [Non-Linguistic Supervision for Contrastive Learning of Sentence Embeddings] (Jian et al, NeurIPS 2022)
>
> [2] [MCSE: Multimodal Contrastive Learning of Sentence Embeddings] (Zhang et al, NAACL-HLT 2022)
>
> ---
>
> We sincerely appreciate the reviewer for their insightful comments and questions.
>
> Best regards,
>
> The Authors

---

### Official Review · Reviewer_rnBv · 2023-08-02

**Soundness:** 4

**Excitement:**

3: Ambivalent: It has merits (e.g., it reports state-of-the-art results, the idea is nice), but there are key weaknesses (e.g., it describes incremental work), and it can significantly benefit from another round of revision. However, I won't object to accepting it if my co-reviewers champion it.

**Paper Topic And Main Contributions:**

The authors of the paper observe that sentence embeddings produced by BERT exhibit a bias towards uninformative words, which consequently hinders their performance on semantic textual similarity (STS) tasks. The goal of the paper is to address this bias and enhance the quality of sentence representations. The paper first analyzes BERT sentence embedding between BERT and SBERT, and then the paper proposes an unsupervised approach called "Diagonal Attention Pooling" (Ditto) to mitigate the anisotropy problem in sentence embeddings without requiring further fine-tuning or additional parameters.

**Reasons To Accept:**

1. The paper analyze the sentence embedding from two perspective with visualized illustration which is impressive and inspiring.
2.  The proposed solution, Diagonal Attention Pooling (Ditto), is simple to understand and implement. It doesn't require any additional training, fine-tuning, or changes to the original model's parameters.
3. The paper backs up its claims with ample empirical evaluations. And the paper is easy to understand.

**Reasons To Reject:**

1. In Ditto, the authors proposed two methods, one utilizing static word embeddings and the other employing diagonal values from the attention matrix. However, the correspondence between the first method and observation 1 needs to be further elucidated.
2. In the experimental section, for both the "Learning-Free Methods" section of Table 1 and the BERT section of Table 2, the authors' approach did not surpass the "Manual Prompt" baseline, which need further solutions or explainations.
3. Since Ditto cannot be applied to methods like SimCSE that use [CLS] for embedding, this may imply a limited applicability scope for Ditto.

**Reproducibility:**

4: Could mostly reproduce the results, but there may be some variation because of sample variance or minor variations in their interpretation of the protocol or method.

**Reviewer Confidence:**

2: Willing to defend my evaluation, but it is fairly likely that I missed some details, didn't understand some central points, or can't be sure about the novelty of the work.

---

> ### Author Rebuttal · Authors · 2023-08-28
>
> We would like to express our gratitude for considering our analysis with visualized illustrations impressive and inspiring, and for finding our claims with ample empirical evaluation easy to understand. We have carefully considered your questions and concerns, and below are our responses to all of your questions.
>
> ---
>
> **Q1: In Ditto, the authors proposed two methods, one utilizing static word embeddings and the other employing diagonal values from the attention matrix. However, the correspondence between the first method and observation 1 needs to be further elucidated.**
>
> **Response:**
> We would like to clarify that we only proposed the novel learning-free method based on Diagonal Attention Pooling (Line 008-014, Line 220-230). The static avg., which is based on averaging the static word embeddings as sentence embeddings, is the baseline we compare to and was introduced in earlier work [1].
>
> [1] [Prompt-bert: Improving BERT sentence embeddings with prompts] (Jiang et al., EMNLP 2022)
>
> ---
>
> **Q2: In the experimental section, for both the "Learning-Free Methods" section of Table 1 and the BERT section of Table 2, the authors' approach did not surpass the "Manual Prompt" baseline, which need further solutions or explainations.**
>
> **Response:**
> We have clarified the drawbacks of the manual prompt approach, which we describe in Line 065-072 of our paper. While it is true that in both the "Learning-Free Methods" section of Table 1 and the BERT section of Table 2, Ditto did not surpass the manual prompt baseline, we already explicitly mentioned in Line 249-252 that the performance of the manual prompt approach can vary significantly depending on the templates used, with scores ranging from 39.34 to 73.44 on the STS-B dev set [1]. In contrast, as shown in Line 252-255, **when applying Ditto to static avg., last avg., and first-last avg., we achieved absolute gains of +3.05, +9.20, and +8.07 on the Avg. score, respectively**. Additionally, as shown in Line 285-295, our Ditto consistently performs well on different pre-trained language models (PLMs) (including RoBERTa and ELECTRA) and outperforms the last manual prompt approach.
>
> [1] [Prompt-bert: Improving BERT sentence embeddings with prompts] (Jiang et al., EMNLP 2022)
>
> ---
>
> **Q3: Since Ditto cannot be applied to methods like SimCSE that use [CLS] for embedding, this may imply a limited applicability scope for Ditto.**
>
> **Response:**
> It is true that Ditto is not directly applicable to sentence representation models such as SimCSE that use [CLS] as sentence embeddings (Line 281-284). However, many strong unsupervised and supervised sentence representation models are based on weighted averages of word representations. Table 1 shows that supervised SBERT first-last avg. significantly outperforms supervised SimCSE, yet Ditto still achieves an absolute gain of 0.17 (84.94 $\rightarrow$ 85.11). This demonstrates that Ditto can improve strong supervised sentence embedding methods as well.
>
> ---
>
> Once again, we sincerely appreciate your insightful comments and questions.
>
> Best regards,
>
> The Authors

---

### Official Review · Reviewer_rde8 · 2023-08-06

**Soundness:** 4

**Excitement:**

4: Strong: This paper deepens the understanding of some phenomenon or lowers the barriers to an existing research direction.

**Paper Topic And Main Contributions:**

the submission proposed a learning-free approach to obtain sentence representations from a pre-trained encoder-only model. The approach is well-motivated by the correspondence between the diagonal entries of the attention mechanism and TF-IDF weights. Moreover, the proposed approach improves the performance of the baseline on STS tasks.

**Reasons To Accept:**

1. the approach is well-motivated using the correlation between attention maps and TF-IDF weights.

2. the approach is indeed learning-free, which can be easily plugged on top of any pre-trained language models with transformers.

3. the performance improvement is solid.

**Reasons To Reject:**

1. TF-IDF itself has certain drawbacks, and simply looking at the linear relationship between sentence representations and TF-IDF may introduce TF-IDF's issues into the resulting sentence representations.

2. I wonder if it would be better to concatenate the avged embedding of the first layer and that of the last layer. The process of averaging them and then taking the cosine similarity creates dot-product terms between the first layer and the last layer, which may not be desirable.

3. there are several prior work on learning-free or parameter-free postprocessing for improving sentence embeddings, and I recommend the authors discuss them or perhaps compare the proposed approach to them.

[1] Jiaqi Mu, Suma Bhat, and Pramod Viswanath. All-but-the-top: Simple and effective postprocessing for word representations. In International Conference on Learning Representations, 2018.

[2] Sanjeev Arora, Yingyu Liang, and Tengyu Ma. A simple but tough-to-beat baseline for sentence
embeddings. In International Conference on Learning Representations, 2017.

**Reproducibility:**

4: Could mostly reproduce the results, but there may be some variation because of sample variance or minor variations in their interpretation of the protocol or method.

**Reviewer Confidence:**

4: Quite sure. I tried to check the important points carefully. It's unlikely, though conceivable, that I missed something that should affect my ratings.

---

> ### Author Rebuttal · Authors · 2023-08-28
>
> We would like to express our gratitude for your appreciation of our approach and for considering it well-motivated, learning-free, easy to plug in, and with solid performance improvement. In the following, we provide responses to all of your questions.
>
> ---
>
> **Q1: TF-IDF itself has certain drawbacks, and simply looking at the linear relationship between sentence representations and TF-IDF may introduce TF-IDF’s issues into the resulting sentence representations.**
>
> **Response:**
> We acknowledge the limitations of TF-IDF, such as the requirement of a large corpus to learn reliable TF-IDF weights. We have discussed these issues in our paper, and compared the performance of first-last TF-IDF with our learning-free Ditto approach (Line 261-270). With reliable TF-IDF computations (Line 261-266), they can indicate word importance. Thus, our approach of studying the correlations between the impact matrix and TF-IDF weights is reasonable.
>
> ---
>
> **Q2: I wonder if it would be better to concatenate the avged embedding of the first layer and that of the last layer. The process of averaging them and then taking the cosine similarity creates dot-product terms between the first layer and the last layer, which may not be desirable.**
>
> **Response:**
> The average embedding of the first and last layer is a common baseline in previous works [1][2]. We have also experimented with concatenating the averaged embeddings of the first and last layer for BERT, and the average score on STS is 55.35, which is **worse than the average score of 56.70 achieved by averaging the averaged embeddings of the first and last layer as reported in the paper**. Additionally, concatenating the hidden states of the first and last layer will double the size and may not necessarily represent the sentence accurately.
>
> [1] [Prompt-bert: Improving BERT sentence embeddings with prompts] (Jiang et al., EMNLP 2022)
>
> [2] [Simcse: Simple contrastive learning of sentence embeddings] (Gao et al., EMNLP 2021)
>
> ---
>
> **Q3: There are several prior work on learning-free or parameter-free postprocessing for improving sentence embeddings, and I recommend the authors discuss them or perhaps compare the proposed approach to them.
> [1] Jiaqi Mu, Suma Bhat, and Pramod Viswanath. All-but-the-top: Simple and effective postprocessing for word representations. In International Conference on Learning Representations, 2018.
> [2] Sanjeev Arora, Yingyu Liang, and Tengyu Ma. A simple but tough-to-beat baseline for sentence embeddings. In International Conference on Learning Representations, 2017.**
>
> **Response:**
> We appreciate the references you provided on parameter-free postprocessing for improving sentence embeddings. We have indeed considered these works and found them not directly related to our work in terms of the specific focus and contributions that our work presents.
>
> Regarding "All-but-the-top: Simple and effective postprocessing for word representations" by Mu et al., their method is designed for static word embeddings such as WORD2VEC and GLOVE. It aims to eliminate the common mean vector and a few top dominating directions from the embeddings. However, these embeddings are not learning-free since the common mean vector and PCA need to be computed on a corpus. This method shares a similar idea with the BERT-whitening approach, which we discussed in our paper (Line 082-085).
>
> Regarding "A simple but tough-to-beat baseline for sentence embeddings" by Arora et al., their method represents the sentence by a weighted average of static word vectors and modifies it using PCA/SVD. Consequently, this method is also designed for static word embeddings and not for BERT-like sentence embeddings. Moreover, it requires calculations of weights (called smooth inverse frequency, SIF) and PCA/SVD, making it not entirely learning-free. Additionally, the SIF reweighting resembles TF-IDF reweighting, which we have already discussed and compared in our paper (Table 1 and Line 261-270).
>
> ---
>
> Once again, we sincerely appreciate your insightful comments and questions.
>
> Best regards,
>
> The Authors

---

### Meta-Review · Area_Chair_qsaH · 2023-09-19

**Recommendation:** 4

**Metareview:**

This paper proposes a method called Diagonal Attention Pooling, an unsupervised way to improve sentence embedding quality. The reviewers converge toward excitement, posing only minor issues regarding clarification and suggestions for future work.

---

### Decision · Program_Chairs · 2023-10-07

**Decision:**

Accept-Main

**Comment:**

This paper proposes a method called Diagonal Attention Pooling, an unsupervised way to improve sentence embedding quality. The reviewers converge toward excitement, posing only minor issues regarding clarification and suggestions for future work.